# Origin of imported SARS-CoV-2 strains in The Gambia identified from whole genome sequences

**Abdoulie Kanteh**[1], **Jarra Manneh**[1], **Sona Jabang**[1], **Mariama A. Kujabi**[1],
**Bakary Sanyang**[1], **Mary A. Oboh**[2], **Abdoulie Bojang**[2], **Haruna S. Jallow**[3],
**Davis Nwakanma**[4], **Ousman Secka**[4], **Anna Roca**[2], **Alfred Amambua-Ngwa**[2],
**Martin Antonio**[5], **Ignatius Baldeh**[3], **Karen Forrest**[6], **Ahmadou Lamin Samateh**[7],
**Umberto D'Alessandro**[2‡], **Abdul Karim Sesay**[1‡*]

**1** Genomics Core Facility, Lab Services, MRCG at LSHTM, Fajara, The Gambia, **2** Disease Control and Elimination, MRCG at LSHTM, Fajara, The Gambia, **3** National Public Health, Laboratories, Kotu, The Gambia, **4** Laboratory Services, MRCG at LSHTM, Fajara, The Gambia, **5** West Africa Platform, MRCG at LSHTM, Fajara, The Gambia, **6** Clinical Service Department, MRCG at LSHTM, Fajara, The Gambia, **7** Ministry of Health, Banjul, The Gambia

‡ These authors are joint senior authors on this work.
* Abdul.Sesay@lshtm.ac.uk

**Data Availability Statement:** The data from the genomes sequenced in the Gambia were submitted and available in GenBank with the following

## Abstract

The SARS-CoV-2 disease, first detected in Wuhan, China, in December 2019 has become a global pandemic and is causing an unprecedented burden on health care systems and the economy globally. While the travel history of index cases may suggest the origin of infection, phylogenetic analysis of isolated strains from these cases and contacts will increase the understanding and link between local transmission and other global populations. The objective of this analysis was to provide genomic data on the first six cases of SARS-CoV-2 in The Gambia and to determine the source of infection. This ultimately provide baseline data for subsequent local transmission and contribute genomic diversity information towards local and global data. Our analysis has shown that the SARS-CoV-2 virus identified in The Gambia are of European and Asian origin and sequenced data matched patients' travel history. In addition, we were able to show that two COVID-19 positive cases travelling in the same flight had different strains of SARS-CoV-2. Although whole genome sequencing (WGS) data is still limited in sub-Saharan Africa, this approach has proven to be a highly sensitive, specific and confirmatory tool for SARS-CoV-2 detection.

## Introduction

The emergence and re-emergence of pathogens such as severe acute respiratory syndrome-coronavirus 2 (SARS-CoV-2) pose a grave threat to human health [1] The SARS-CoV-2 disease, first detected in Wuhan, China, in December 2019 has become a global pandemic [2] and is causing an unprecedented burden on health care systems and the economy globally [3–6]. The estimated number of cases has increased exponentially [6], especially in Europe and

accession numbers: MZ040125, MZ040126, MZ040127, MZ040128, MZ040129, MZ040130).

**Funding:** The author(s) received no specific funding for this work.

**Competing interests:** The authors have declared that no competing interests exist.

America, with significant but variable case-fatality rates inter-continentally. By April 27[th], 2021, there were over 148 million SARS-CoV-2 confirm cases and 3.12 million deaths globally [7]. The Gambia, a tourism hotspot, reported more than five thousand SARS-Cov-2 cases and 173 deaths [7].

While the travel history of index cases may suggest the origin of infection, phylogenetic analysis of isolated strains from these cases and contacts will increase the understanding and link between local transmission and imported strains. The phylogenetic analyses of global SARS-CoV-2 sequences provide insight into the relatedness of strains from different areas and suggest the transmission of four super-clades, [8] geographically clustering into viral isolates from Asia (China), US (two super clades) and Europe. SARS-CoV-2 pose significant risk to global health, hence the need for effective strategies to detect the sources of infections, outbreaks and transmission patterns in different geographical settings. The objective of this analysis was to provide genomic data on the first six cases of SARS-CoV-2 in The Gambia and to determine the source of the strains. This would ultimately provide baseline data for monitoring subsequent local transmission(s) and contribute genomic diversity information towards local and global data.

## Materials and methods

The Scientific Coordinating Committee at Medical Research Council Unit The Gambia at LSHTM has approved the manuscript to be publish given that Ethics Committee approval is not required to publish routine public health data.

Oxford Nanopore (GridION) and Illumina (MiSeq) platforms were used to sequence the viral genomes as summarised in Table 1. Total RNA was purified from Nasopharyngeal-Oropharyngeal (NP-OP) swabs using QiaAmp viral RNA mini kit (Qiagen– 52906). Library preparation was done following the ARTIC sequencing protocol [9] with little modification for Illumina sequencing. FASTQ files were analysed following the ARTIC bioinformatics pipeline [9] to generate consensus sequences. Phylogenetic tree was constructed using IQTREE (v1.3.11.1) [10] and visualised and annotated using Interactive Tree of Life (ITOL) (v5) [11].

**Table 1. Sample information for COVID-19 sequenced cases from The Gambia.**

| Case ID | Travelled from | Date Reported | Current Status | Number of samples submitted | Time points | Library prep type | | | Sequencing | |
|---|---|---|---|---|---|---|---|---|---|---|
| | | | | | | Depletion | ARTIC amplicon (NEB) | ARTIC amplicon (ONT -LSK109) | Illumina (MiSeq) | Nanopore (GridION) |
| A | London | 16/03/20 | Recovered | 4 | Days 0,4,7,10 | 2 | 4 | 4 | 4 | 4 |
| B | Bangladesh | 19/03/20 | Dead | 1 | Day 0 | 0 | 1 | 1 | 1 | 1 |
| C | France | 20/03/20 | Recovered | 1 | Day 0, | 0 | 1 | 1 | 1 | 1 |
| D | France | 26/03/20 | Active | 2 | Day 0,11 | 0 | 1 | 2 | 1 | 2 |
| E | Netherland | 23/03/20 | Active | 2 | Day 0,14 | 0 | 1 | 2 | 1 | 2 |
| F | Italy | 13/03/20 | Recovered | 1 | Day 0 | 0 | 1 | 1 | 1 | 1 |
| Total | | | | 11 | | 2 | 9 | 11 | 9 | 11 |

Cases A-D = Confirmed RT-PCR COVID-19 cases.

Case E = Indeterminate by RT-PCR.

Case F = RT-PCR COVID-19 negative.

Cases A and D travelled to The Gambia in the same flight.

Cases C and D both travelled from France.

Note: Status of patients given here was at the time of sequencing.

Selection of genomes from GISAID for comparison with the isolated strains was based on patients' travel history and the major geographical spread of the pandemic.

## Results and discussion

Whole genome sequencing data was generated from the first six confirmed cases in the Gambia to determine the source of these strains as well as provide baseline data for subsequent local transmission. We also sought to assess if Nanopore sequencing platform, a portable and cheap sequencer can produce high quality genomic data with accurate SNPs for phylogenetic inference when compared to Illumina.

Our analysis showed that six genomes (4 samples from Case A, 1 from case C and 1 from case D) from the Gambian samples clustered with the European (Spanish and United Kingdom) SARS-CoV-2 strains Fig 1. This correlates with the patients' travel history as they had been in Europe before arriving in The Gambia. Strains from cases C and D, both of whom travelled from France, were more closely related to the Spanish strain. Interestingly, cases D and A travelled to The Gambia on the same flight, however, their strains clustered on different nodes, indicating that they could have been infected independently, before the start of their journeys. Case F was a negative SARS-CoV-2 sample thus excluded from the phylogenetic analysis. Although only few genomes were selected elsewhere for our phylogenetic inference as shown in Fig 1, similar topology was observed when genomes deposited on Nextstrain [12] a public repository for SARS-CoV-2 sequences were included as shown in Fig 2.

Although viruses are known to mutate and change rapidly [13], the viral genome of case A collected at different time-points clustered on the same node indicating the patient had been shedding the same virus with no observed polymorphism according to our Nanopore data. The same sample sequenced on the MiSeq resulted in a longer branch length at day10 when compared to other time points. SNP analysis showed seven more SNPs on day10 sample compared to the same sample sequenced on the MinION (Nanopore data). This was an interesting finding, however, the SNP winked by the Nanopore, might be due to higher accuracy on

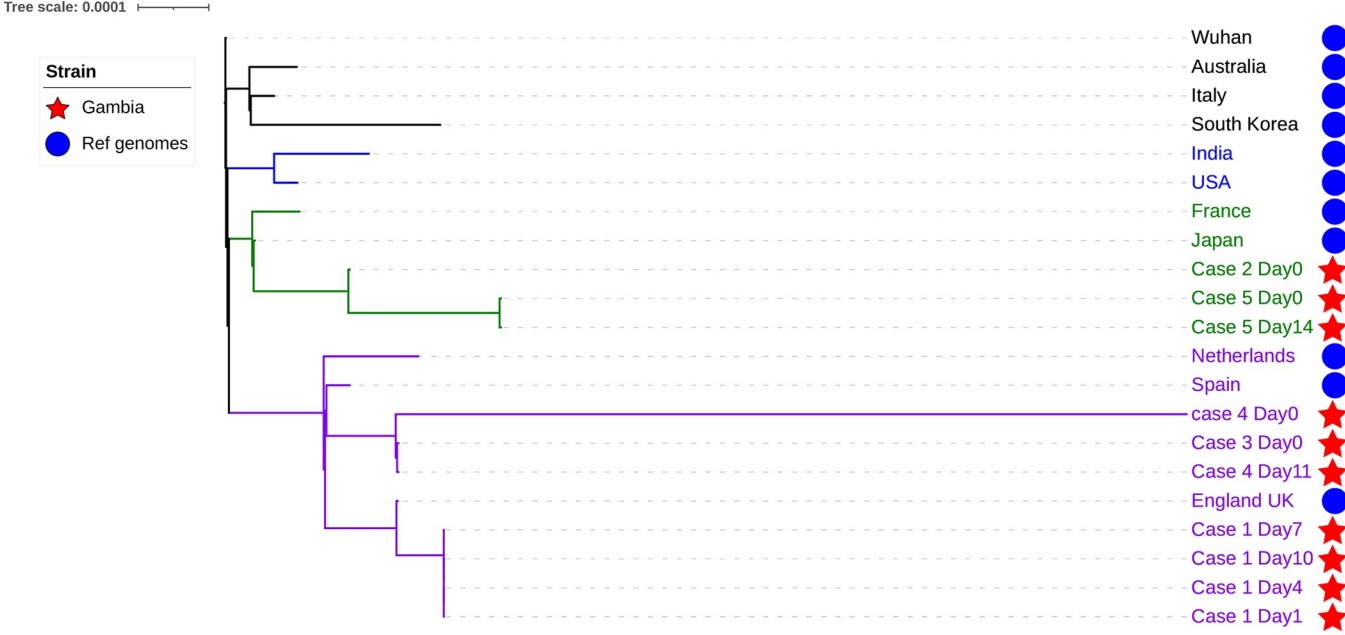

**Fig 1. A maximum likelihood phylogenetic tree of Gambia SARS-CoV-2 genomes and 11 SARS-CoV-2 strains isolated in different parts of the world.**

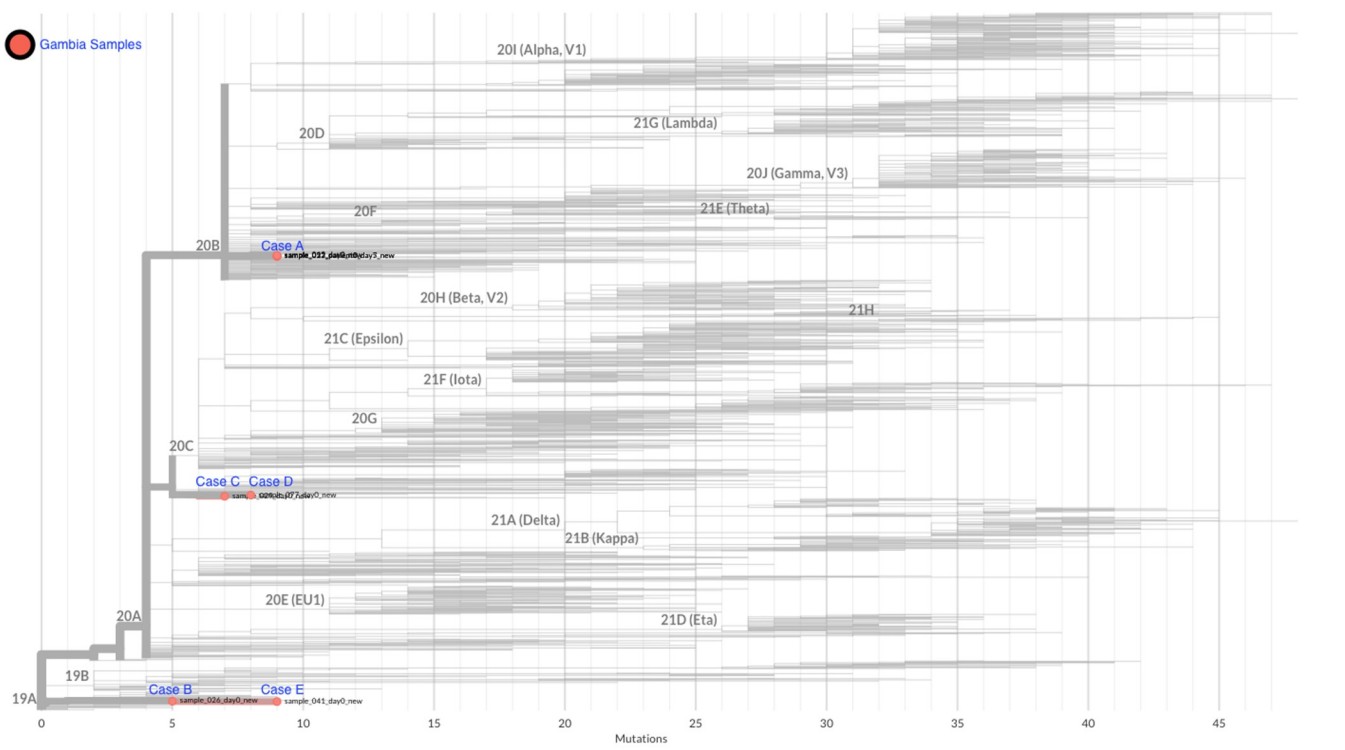

**Fig 2. A maximum likelihood phylogenetic tree of Gambia SARS-CoV-2 genomes isolated from The Gambia and those found elsewhere.** The tree was constructed using Nextclade.

Illumina or perhaps might as well be a sequencing error. This was difficult to ascertain give that the sample was run only once using both platforms.

The viral genome from case B who initiated travel from Bangladesh and then across four other countries, including Senegal, before arriving in The Gambia, clustered with a strain from Japan. This case may have contracted the virus in Asia and his travel history suggests he could have contributed to infections in other countries. The two isolates from case E at different time points clustered with strains from Japan as well. Interestingly, case E samples were indeterminate by rRT-PCR diagnostics. The indeterminate diagnostic rRT-PCR result could be due to low sensitivity of the assay, an indication of low viral density of SARS-CoV-2 in the sample. Therefore, subsequent follow up for such indeterminate cases is imperative for accurate information and aid our understanding of the disease progression as well as the evolution of this novel virus strain under different case management environments.

## Summary

Our analysis has shown that the SARS-CoV-2 strains identified in The Gambia are of European and Asian origin and sequence data matched patients' travel history. In addition, we were able to show that two COVID-19 positive cases travelling in the same flight had different strains of SARS-CoV-2. Although whole genome sequence (WGS) data of SARS-Cov-2 in sub-Saharan Africa is still limited, this approach has proven to be a highly sensitive, specific and confirmatory tool for SARS-CoV-2 detection. Hence, the use of second and third generation sequencing technologies coupled with bioinformatics will convincingly provide data for monitoring transmission dynamics.

We have also demonstrated that the Nanopore platform with its flexibility for number of samples per run, and the generation of data in real-time and at a reasonable cost makes it most suitable for outbreak monitoring, especially in resource limited areas. This would go a long way in providing knowledge on the molecular epidemiology of this disease, give the true burden of the disease in resource limited setting as well as provide information for African specific mutations which could have a significant implication on vaccine development and roll out.

## Acknowledgments

We acknowledge the use of CLIMB server for the cloud-based analysis, the field sample collection by the teams at Ministry of Health, Epidemiology Department, Thushan de Silva for helpful discussion on ARTIC protocol and sequencing, Covid-19 laboratory diagnostic staff, and at MRCG at LSHTM Logistics, Staff at CSD, COVID-19 Emergency Management.

## Author Contributions

**Conceptualization:** Abdoulie Kanteh, Jarra Manneh.

**Formal analysis:** Abdoulie Kanteh.

**Writing – original draft:** Abdoulie Kanteh, Jarra Manneh, Sona Jabang, Mariama A. Kujabi, Bakary Sanyang, Mary A. Oboh, Abdoulie Bojang.

**Writing – review & editing:** Abdoulie Kanteh, Jarra Manneh, Sona Jabang, Mariama A. Kujabi, Bakary Sanyang, Mary A. Oboh, Abdoulie Bojang, Haruna S. Jallow, Davis Nwakanma, Ousman Secka, Anna Roca, Alfred Amambua-Ngwa, Martin Antonio, Ignatius Baldeh, Karen Forrest, Ahmadou Lamin Samateh, Umberto D'Alessandro, Abdul Karim Sesay.

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
