## [Decision Letter · Decision Letter 0]

6 Jan 2021

PONE-D-20-33232

Origin of imported SARS-CoV-2 strains in The Gambia identified from whole genome sequences

PLOS ONE

Dear Dr. Kanteh,

Thank you for submitting your manuscript to PLOS ONE. After careful consideration, we feel that it has merit but does not fully meet PLOS ONE’s publication criteria as it currently stands. Therefore, we invite you to submit a revised version of the manuscript that addresses the points raised during the review process.

Your manuscript was reviewed by 4 experts in the field. They identified many important problems in your submission. Please carefully consider the attached comments and provide point-by-point responses.

We look forward to receiving your revised manuscript.

Kind regards,

Yury E Khudyakov, PhD

Academic Editor

PLOS ONE

Journal Requirements:

2. We note that you are reporting an analysis of a microarray, next-generation sequencing, or deep sequencing data set. PLOS requires that authors comply with field-specific standards for preparation, recording, and deposition of data in repositories appropriate to their field. Please upload these data to a stable, public repository (such as ArrayExpress, Gene Expression Omnibus (GEO), DNA Data Bank of Japan (DDBJ), NCBI GenBank, NCBI Sequence Read Archive, or EMBL Nucleotide Sequence Database (ENA)). In your revised cover letter, please provide the relevant accession numbers that may be used to access these data. For a full list of recommended repositories, see http://journals.plos.org/plosone/s/data-availability#loc-omics or http://journals.plos.org/plosone/s/data-availability#loc-sequencing

3. Please provide additional details regarding participant consent.

In the ethics statement in the Methods and online submission information, please ensure that you have specified (i) whether consent was informed and (ii) what type you obtained (for instance, written or verbal, and if verbal, how it was documented and witnessed). If your study included minors, state whether you obtained consent from parents or guardians. If the need for consent was waived by the ethics committee, please include this information.

Reviewers' comments:

Reviewer's Responses to Questions

**Comments to the Author**

1. Is the manuscript technically sound, and do the data support the conclusions?

Reviewer #1: Yes

Reviewer #2: Partly

Reviewer #3: Partly

Reviewer #4: Yes

2. Has the statistical analysis been performed appropriately and rigorously? 

Reviewer #1: Yes

Reviewer #2: N/A

Reviewer #3: No

Reviewer #4: N/A

3. Have the authors made all data underlying the findings in their manuscript fully available?

Reviewer #1: Yes

Reviewer #2: Yes

Reviewer #3: No

Reviewer #4: No

4. Is the manuscript presented in an intelligible fashion and written in standard English?

Reviewer #1: Yes

Reviewer #2: Yes

Reviewer #3: Yes

Reviewer #4: Yes

5. Review Comments to the Author

Reviewer #1: Dear author(s),

I have read your submission and it was interesting. Only there are minor things that you should consider for having better insight to your study. You may pay attention to this point that whether or not this study has been done previously by others and compare your results. Another point that you should consider is that what is your main goal for future of performing this work and what was your expectation out of that data.

Reviewer #2: In this paper, Kanteh et al. reported whole genome sequence of SARS-CoV-2 strains detected in Gambia and their phylogenetic analysis with other strains outside of the country. Given little viral sequence information from African countries, the study provides us with a clue to elucidate the global transmission dynamics of the virus at the relatively early time point of the current outbreak. However, I have several concerns in the manuscript to be addressed before the publication.

Major points

1. Absolute “origin” or “source” cannot be inferred from phylogenetic tree. That is because there must be other viral strains from other countries that are more closely related to Gambian strains but unsampled and not included in their phylogenetic analysis.

2. Related to the point #1, viral sequences from other countries included in the phylogenetic analysis are very few. There should be many more available sequence data of viral strains collected by March 2020 from all over the world in GISAID, ViPR, and so on.

3. Difference in sequences between Illumina and Nanopre protocols should be done by not phylogenetic trees but direct comparison of SNPs (number and positions) between them.

4. Related to the point #3, the authors had better show either of only Figure 2 or Figure 3.

Minor points

5. There are a lot of typo in the manuscript. Please read the manuscript carefully and correct them before re-submission.

6. (Lines 34 and 203) “Wuhan reference genome” is not a scientific term. Please indicate the actual strain name as described in the line 177.

7. Texts in the lines 76 and 77-78 are redundant.

8. The names of genes for the screening and confirmation should be specified here (as described in the Table 2.)

9. I personally think Figure 1 is unnecessary and can be removed from the manuscript. Yet, I will leave the decision to the editor and authors.

10. In the table 2, there is “Ct” in the third column title. However, I do not see any Ct values in the table.

11. (Line 249) WGS should be spelled out.

Reviewer #3: The authors conducted complete genomic sequencing of the first 6 cases of SARS-CoV-2 in The Gambia to determine/confirm their origin/genetic characteristics using Illumina MiSeq and Nanopore platforms. The study addresses an important issue because the knowledge on the genetic characteristic of SARS-CoV-2 strains circulating in The Gambia/Adrica. The study also appears to be technically sound.

The results presented/discussed are insufficient for a full-length research article. If no additional analysis (see below) is done, the submission must be converted to a short communication.

Overall, the manuscript is too focused on technical details of both sequencing methods employed and presents very little detail on the genomic/molecular analysis of the Gambian SARS-CoV-2 strains. This has to be changed by mostly re-writing the manuscript.

It is not clear why the two sequencing approaches had to be used: Illumina MiSeq and Nanopore. Although, the authors stated that both methods had their own advantages, it is not clear why the two methods were needed to just determine the strain origin. “While Illumina sequencing may be more accurate in determining within sample-diversity, Nanopore data can help with the understanding of the linkage between SNPs within individual virions”. The importance/relevance of determining within host/virion diversity in this study is not clear. It is well established fact that coronaviruses exist as quasispecies within the same host.

The abstract needs to be re-written. Currently, it’s a mere statement of the methodology used in the study. It has to be re-written to properly convey the problem/background, approach, findings and conclusion of the study.

Accession numbers for the newly generated sequences must be provided, but were missing.

Issues with the analysis:

More SARS-CoV-2 strains should be included in the phylogenetic trees.

Phylogenetic distance and bootstrap values should be reflected on the trees.

The meaning of the dashed lines should be explained in the figure legends.

Recombination analysis should be conducted.

Reviewer #4: This is a nice short paper on sequencing some SARS-CoV-2 genomes from Gambia. their methods seem fine, and the sequences seem reasonable. My only concern is that they are not deposited to GenBank. I can't access them from GISAID without permissions to their web pages.

6. PLOS authors have the option to publish the peer review history of their article (what does this mean?). If published, this will include your full peer review and any attached files.

Reviewer #1: **Yes: **Leila Mousavizadeh

Reviewer #2: No

Reviewer #3: **Yes: **Anastasia Vlasova

Reviewer #4: No

---

## [Author Response · Author response to Decision Letter 0]

18 May 2021

Medical Research Council Unit The Gambia at LSHTM

 P.O Box 273

 Banjul, The Gambia

Dear sir,

I wish to thank you for your time and effort in reviewing our manuscript title “Origin of imported SARS-CoV-2 strains in The Gambia identified from whole genome sequences submitted in your Journal. The authors have thoroughly thought about all the major comments made on the manuscript and decided to change the paper to a short communication. Please find below answers (in red) to the concerns highlighted by the reviewers. 

Journal Requirements:

 I have formatted the manuscript using the guidelines highlighted above. 

2. We note that you are reporting an analysis of a microarray, next-generation sequencing, or deep sequencing data set. PLOS requires that authors comply with field-specific standards for preparation, recording, and deposition of data in repositories appropriate to their field. Please upload these data to a stable, public repository (such as ArrayExpress, Gene Expression Omnibus (GEO), DNA Data Bank of Japan (DDBJ), NCBI GenBank, NCBI Sequence Read Archive, or EMBL Nucleotide Sequence Database (ENA)). In your revised cover letter, please provide the relevant accession numbers that may be used to access these data. For a full list of recommended repositories, see http://journals.plos.org/plosone/s/data-availability#loc-omics or http://journals.plos.org/plosone/s/data-availability#loc-sequencing

The genomes are now deposited at GenBank with the following submission number: SUB9545765 

3. Please provide additional details regarding participant consent.

In the ethics statement in the Methods and online submission information, please ensure that you have specified (i) whether consent was informed and (ii) what type you obtained (for instance, written or verbal, and if verbal, how it was documented and witnessed). If your study included minors, state whether you obtained consent from parents or guardians. If the need for consent was waived by the ethics committee, please include this information.

 This is sorted accordingly 

Reviewers' comments:

Reviewer's Responses to Questions

Comments to the Author

1. Is the manuscript technically sound, and do the data support the conclusions?

Reviewer #1: Yes

Reviewer #2: Partly

Reviewer #3: Partly

Reviewer #4: Yes

2. Has the statistical analysis been performed appropriately and rigorously? 

Reviewer #1: Yes

Reviewer #2: N/A

Reviewer #3: No

Reviewer #4: N/A

3. Have the authors made all data underlying the findings in their manuscript fully available?

Reviewer #1: Yes

Reviewer #2: Yes

Reviewer #3: No

Reviewer #4: No

4. Is the manuscript presented in an intelligible fashion and written in standard English?

Reviewer #1: Yes

Reviewer #2: Yes

Reviewer #3: Yes

Reviewer #4: Yes

5. Review Comments to the Author

Reviewer #1: Dear author(s),

I have read your submission and it was interesting. Only there are minor things that you should consider for having better insight to your study. You may pay attention to this point that whether or not this study has been done previously by others and compare your results. Another point that you should consider is that what is your main goal for future of performing this work and what was your expectation out of that data.

Reviewer #2: In this paper, Kanteh et al. reported whole genome sequence of SARS-CoV-2 strains detected in Gambia and their phylogenetic analysis with other strains outside of the country. Given little viral sequence information from African countries, the study provides us with a clue to elucidate the global transmission dynamics of the virus at the relatively early time point of the current outbreak. However, I have several concerns in the manuscript to be addressed before the publication.

Major points

1. Absolute “origin” or “source” cannot be inferred from phylogenetic tree. That is because there must be other viral strains from other countries that are more closely related to Gambian strains but unsampled and not included in their phylogenetic analysis.

The reviewer is very right about this point. However, at the time of analysis, we ran all the sequences against the available genomes on Nextclade (n=4645) and constructed a phylogenetic tree. Despite the number of genomes added, the origin or source still remains the same as seen in the image below. 

2. Related to the point #1, viral sequences from other countries included in the phylogenetic analysis are very few. There should be many more available sequence data of viral strains collected by March 2020 from all over the world in GISAID, ViPR, and so on.

Yes, this is true. We actually ran our samples against other genomes from Nexstrain, however, we still have similar topology. Selection of genomes from GISAID was based on our pateint’s travel history.

3. Difference in sequences between Illumina and Nanopore protocols should be done by not phylogenetic trees but direct comparison of SNPs (number and positions) between them.

I agree with you. However, if there was huge difference in terms of the number of SNPs from each platform, this would perhaps alter the topology of the tree. However, this was not seen except in one sample, where the Illumina data had a longer branch length that the Nanopore. We found that there were more SNPs from the Illumina than the Nanopore data. 

4. Related to the point #3, the authors had better show either of only Figure 2 or Figure 

Given that both trees look similar, we decided to include only one in the manuscript. We also added a phylogenetic tree showing all the genomes in Nextclade between march and April 2020.

Minor points

5. There are a lot of typo in the manuscript. Please read the manuscript carefully and correct them before re-submission.

This has been corrected.

6. (Lines 34 and 203) “Wuhan reference genome” is not a scientific term. Please indicate the actual strain name as described in the line 177.

Okay. This is sorted.

7. Texts in the lines 76 and 77-78 are redundant.

This is sorted accordingly. 

8. The names of genes for the screening and confirmation should be specified here (as described in the Table 2.) 

9. I personally think Figure 1 is unnecessary and can be removed from the manuscript. Yet, I will leave the decision to the editor and authors.

10. In the table 2, there is “Ct” in the third column title. However, I do not see any Ct values in the table.

11. (Line 249) WGS should be spelled out. 

Reviewer #3: The authors conducted complete genomic sequencing of the first 6 cases of SARS-CoV-2 in The Gambia to determine/confirm their origin/genetic characteristics using Illumina MiSeq and Nanopore platforms. The study addresses an important issue because the knowledge on the genetic characteristic of SARS-CoV-2 strains circulating in The Gambia/Adrica. The study also appears to be technically sound.

The results presented/discussed are insufficient for a full-length research article. If no additional analysis (see below) is done, the submission must be converted to a short communication.

Overall, the manuscript is too focused on technical details of both sequencing methods employed and presents very little detail on the genomic/molecular analysis of the Gambian SARS-CoV-2 strains. This has to be changed by mostly re-writing the manuscript.

It is not clear why the two sequencing approaches had to be used: Illumina MiSeq and Nanopore. Although, the authors stated that both methods had their own advantages, it is not clear why the two methods were needed to just determine the strain origin. “While Illumina sequencing may be more accurate in determining within sample-diversity, Nanopore data can help with the understanding of the linkage between SNPs within individual virions”. The importance/relevance of determining within host/virion diversity in this study is not clear. It is well established fact that coronaviruses exist as quasispecies within the same host.

The abstract needs to be re-written. Currently, it’s a mere statement of the methodology used in the study. It has to be re-written to properly convey the problem/background, approach, findings and conclusion of the study.

Accession numbers for the newly generated sequences must be provided, but were missing.

Issues with the analysis:

More SARS-CoV-2 strains should be included in the phylogenetic trees.

Phylogenetic distance and bootstrap values should be reflected on the trees.

The meaning of the dashed lines should be explained in the figure legends.

Recombination analysis should be conducted.

Reviewer #4: This is a nice short paper on sequencing some SARS-CoV-2 genomes from Gambia. their methods seem fine, and the sequences seem reasonable. My only concern is that they are not deposited to GenBank. I can't access them from GISAID without permissions to their web pages.

The genomes are now available at GenBank with the submission: SUB9545765

6. PLOS authors have the option to publish the peer review history of their article (what does this mean?). If published, this will include your full peer review and any attached files.

Do you want your identity to be public for this peer review? For information about this choice, including consent withdrawal, please see our Privacy Policy.

Reviewer #1: Yes: Leila Mousavizadeh

Reviewer #2: No

Reviewer #3: Yes: Anastasia Vlasova

Reviewer #4: No

---

## [Decision Letter · Decision Letter 1]

28 May 2021

PONE-D-20-33232R1

Origin of imported SARS-CoV-2 strains in The Gambia identified from whole genome sequences

PLOS ONE

Dear Dr. Kanteh,

Thank you for submitting your manuscript to PLOS ONE. After careful consideration, we feel that it has merit but does not fully meet PLOS ONE’s publication criteria as it currently stands. Therefore, we invite you to submit a revised version of the manuscript that addresses the points raised during the review process.

The revised manuscript was reviewed by 2 original reviewers. Both still identified serious problems in your revision.  Please review the attched comments and provide carefully conceived responses.

We look forward to receiving your revised manuscript.

Kind regards,

Yury E Khudyakov, PhD

Academic Editor

PLOS ONE

Reviewers' comments:

Reviewer's Responses to Questions

**Comments to the Author**

1. If the authors have adequately addressed your comments raised in a previous round of review and you feel that this manuscript is now acceptable for publication, you may indicate that here to bypass the “Comments to the Author” section, enter your conflict of interest statement in the “Confidential to Editor” section, and submit your "Accept" recommendation.

Reviewer #2: (No Response)

Reviewer #4: (No Response)

2. Is the manuscript technically sound, and do the data support the conclusions?

Reviewer #2: Partly

Reviewer #4: Yes

3. Has the statistical analysis been performed appropriately and rigorously? 

Reviewer #2: N/A

Reviewer #4: Yes

4. Have the authors made all data underlying the findings in their manuscript fully available?

Reviewer #2: Yes

Reviewer #4: No

5. Is the manuscript presented in an intelligible fashion and written in standard English?

Reviewer #2: Yes

Reviewer #4: Yes

6. Review Comments to the Author

Reviewer #2: The revised manuscript has been improved to be concise and easy to follow. Still, I have three big concerns in the revised manuscript.

Lines 95-99:

Their finding that 7 SNPs that were observed by Illumina but not by Nanopore appeared only after 10 days of infection in the same individual seem surprisingly a big number to me. The authors argued that "The SNP winked by the Nanopore, might be due to higher accuracy on Illumina..." How could the authors exclude the possibility that there was something wrong in the Illumina run for the particular sample?

Figure 1 and texts:

Case F was totally missed in the tree and main texts.

Figure 2 and texts:

Although the authors included and indicated 3 Gambian sequences in the tree, they should include and show the sequences of all 6 cases. Besides, there is no explanation about the figure in the main texts at all.

Reviewer #4: I still can't find the sequences....

"The genomes are now deposited at GenBank with the following submission number: SUB9545765"

SUB9545765 is not an accession number. I went to the NCBI pages, and did a search against all of the databases, and could not find "SUB9545765" in any of their databases.

The NCBI SARS-CoV-2 portal lives here:

https://www.ncbi.nlm.nih.gov/sars-cov-2/

and there's about a half-million genomes in there, and here the accession number usually contains two letters, followed by six numbers - for example: FR988027 or OU032008 are both accession numbers.

Another example, from a whole genome sequencing project looks like this:

WGS of SARS-CoV-2 circulating in Spain

GenBank Accession: ERX5596932

BioProject ID: PRJEB43166

SRA study link: ERP127101

SRA run number: ERR5956411

7. PLOS authors have the option to publish the peer review history of their article (what does this mean?). If published, this will include your full peer review and any attached files.

Reviewer #2: No

Reviewer #4: **Yes: **David Ussery

---

## [Author Response · Author response to Decision Letter 1]

12 Jul 2021

This uploaded in Attach File section

---

## [Decision Letter · Decision Letter 2]

9 Aug 2021

Origin of imported SARS-CoV-2 strains in The Gambia identified from whole genome sequences

PONE-D-20-33232R2

Dear Dr. Kanteh,

We’re pleased to inform you that your manuscript has been judged scientifically suitable for publication and will be formally accepted for publication once it meets all outstanding technical requirements.

Kind regards,

Yury E Khudyakov, PhD

Academic Editor

PLOS ONE

Additional Editor Comments (optional):

Reviewers' comments:

Reviewer's Responses to Questions

**Comments to the Author**

1. If the authors have adequately addressed your comments raised in a previous round of review and you feel that this manuscript is now acceptable for publication, you may indicate that here to bypass the “Comments to the Author” section, enter your conflict of interest statement in the “Confidential to Editor” section, and submit your "Accept" recommendation.

Reviewer #2: All comments have been addressed

2. Is the manuscript technically sound, and do the data support the conclusions?

Reviewer #2: (No Response)

3. Has the statistical analysis been performed appropriately and rigorously? 

Reviewer #2: (No Response)

4. Have the authors made all data underlying the findings in their manuscript fully available?

Reviewer #2: (No Response)

5. Is the manuscript presented in an intelligible fashion and written in standard English?

Reviewer #2: (No Response)

6. Review Comments to the Author

Reviewer #2: (No Response)

7. PLOS authors have the option to publish the peer review history of their article (what does this mean?). If published, this will include your full peer review and any attached files.

Reviewer #2: No

---

## [Editor Report · Acceptance letter]

16 Aug 2021

PONE-D-20-33232R2 

Origin of imported SARS-CoV-2 strains in The Gambia identified from whole genome sequences. 

Dear Dr. Kanteh:

I'm pleased to inform you that your manuscript has been deemed suitable for publication in PLOS ONE. Congratulations! Your manuscript is now with our production department. 

Kind regards, 

on behalf of

Dr. Yury E Khudyakov 

Academic Editor

PLOS ONE